# High Performance Gold Nanorods@DNA Self-Assembled Drug-Loading System for Cancer Thermo-Chemotherapy in the Second Near-Infrared Optical Window

**DOI:** 10.3390/pharmaceutics14051110

**Published:** 2022-05-23

**Authors:** Wei Chang, Junfeng Wang, Jing Zhang, Qing Ling, Yumei Li, Jie Wang

**Affiliations:** 1Inflammation and Immune Mediated Diseases Laboratory of Anhui Province, School of Pharmacy, Anhui Medical University, Hefei 230032, China; changw@bbmc.edu.cn (W.C.); wjf2353258194@outlook.com (J.W.); zhangjing10087@outlook.com (J.Z.); lingqing22luan@163.com (Q.L.); 2Anhui Engineering Technology Research Center of Biochemical Pharmaceuticals, Faculty of Pharmacy, Bengbu Medical College, Bengbu 233030, China; 3School of Basic Medicine, Gannan Medical University, Ganzhou 341000, China

**Keywords:** gold nanorods@CTA, NIR-II window, thermo-chemotherapy, photoacoustic imaging

## Abstract

In terms of synergistic cancer therapy, biological nanomaterials with a second near-infrared (NIR-II) window response can greatly increase photothermal effects and photoacoustic imaging performance. Herein, we report a novel stimuli-responsive multifunctional drug-loading system which was constructed by integrating miniature gold nanorods (GNR) as the NIR-II photothermal nanorods and cyclic ternary aptamer (CTA) composition as a carrier for chemotherapy drugs. In this system, doxorubicin hydrochloride (DOX, a chemotherapy drug) binds to the G-C base pairs of the CTA, which exhibited a controlled release behavior based on the instability of G-C base pairs in the slightly acidic tumor microenvironment. Upon the 1064 nm (NIR-II biowindow) laser irradiation, the strong photothermal and promoted cargo release properties endow gold nanorods@CTA (GNR@CTA) nanoparticles displaying excellent synergistic anti-cancer effect. Moreover, the GNR@CTA of NIR also possesses thermal imaging and photoacoustic (PA) imaging properties due to the strong NIR region absorbance. This work enables to obtaining a stimuli-responsive “all-in-one” nanocarrier, which are promising candidate for bimodal imaging diagnosis and chemo-photothermal synergistic therapy.

## 1. Introduction

Cancer is considered to be one of the main causes of death in humans [1]. Given the high risk and death rate of cancer, researchers around the world have been struggling to develop the more accurate and rapid diagnostic strategies and effective therapies to fight against cancer [2]. The conventional treatment methods, such as surgery, radiotherapy, and chemotherapy, have limitations like serious side effects and unsatisfactory treatment outcomes [3,4,5]. To solve these problems, combination therapy has gained much attention. With the rapid development of nanomaterials, photothermal therapy based on the integration of diagnostic and therapeutic functions holds enormous potential for tumor detection and real-time drug distribution [6,7,8].

As a minimally invasive therapy, photothermal therapy has received extensive attention. It uses photothermal transition agents (PTAs) to generate heat under light irradiation and induce local hyperthermia to trigger the death of cancer cells [9,10]. Among existing PTAs, two distinct spectral ranges NIR-I (650–900 nm) and NIR-II (1000–1350 nm) have been proposed as the most promising PTAs with potential biodegradability, improved biocompatibility, and high reproducibility [11]. In the past few decades, the absorption of PTAs in the NIR-I window has gotten great development, but the clinical application is still not satisfactory. Recently, the higher allowable exposure and deeper tissue penetration of NIR-II window offer advantages in terms of better therapeutic efficacy for tumors versus the NIR-I window [12,13]. Therefore, high quality NIR-II responsive PTAs are worthwhile to be developed. However, one major obstacle, to date, is the limited choice of contrast agents in this spectral range. Common PTAs, including organic dyes [14,15], carbon nanomaterials [16,17], inorganic semiconductor materials [18,19], and precious metal materials [20,21], in this window are largely uncharacterized for their cytotoxicity. Among them, AuNPs, a precious metal material, have received widespread attention based on their outstanding physical and chemical properties, good biocompatibility, and excellent local surface plasmon resonance (LSPR) properties. By changing its shape, size, and shape, AuNPs can extend the LSPR spectrum from the visible region to the near-infrared region (650–1100 nm) [22,23,24]. Usually, AuNPs are used into NIR-I windows, but rarely for the NIR-II biological window [25]. Generally speaking, in order to pursue high aspect ratios to achieve the photothermal response of the NIR-II window, the nanometer length of the gold nano particle can exceed 100 nm, even reach more than 150 nm, which severely restricts the biological application of the gold nano particle, especially in vivo applications. The critical size of nanoparticles is limited to ~100 nm, beyond this range, the penetration of nanoparticles into the tumor is limited [26,27].

Based on the above situation, we reasonably designed and successfully constructed a gold nanorods (GNR) with a high aspect ratio in a limited particle size range to achieve a superior photothermal response for the NIR-II window. The new gold nanorods has a particle size of about 65 nm and an aspect ratio of 8 ± 2. The UV-visible-NIR spectrum shows that the absorption of this GNR is 1050 nm. It has excellent photothermal response of the NIR-II window. We bonded CTA to the surface of GNR and constructed a nano-assembly GNR@CTA that specifically targets the tumor site. On the one hand, CTA can select specific tumor sites to target, and at the same time, the abundant G-C base pairs provide a large number of drug loading sites for DOX. On the other hand, GNR can realize the photothermal response treatment of NIR-II window and photoacoustic imaging of tumor sites, and at the same time, the photothermal effect also further promotes the rupture of G-C base pairs in CTA, leading to the accelerated release of DOX. The experimental results of in vitro and in vivo indicate that nanocomposite materials have unique biocompatibility and synergistically enhanced anti-tumor effects compared with photothermal or chemotherapy alone. 

## 2. Materials and Methods

### 2.1. Materials and Reagents

All chemicals in this study were used as received: cetyl trimethyl ammonium bromide (CTAB, Sigma-Aldrich, Shanghai, China), gold (III) chloride hydrate (HAuCl_4_, Sigma-Aldrich), sodium borohydride (NaBH_4_, Sigma-Aldrich), silver nitrate (AgNO_3_, Sigma-Aldrich), adriamycin (Beijing Dingguo Changsheng Biotechnology Co., Ltd., Beijing, China), hydroquinone (Sigma-Aldrich), hydrochloric acid (HCl, Sigma-Aldrich), sodium hydroxide (1N solution, NaOH, Sinopharm Chemical Reagent Co., Ltd., Beijing, China), dulbecco’s modified eagle medium (DMEM, HyClone), and apoptosis detection kit (Beijing Dingguo Changsheng Biotechnology Co., Ltd.).

The main instruments, models and manufacturers used in this chapter: CO_2_ laboratory incubator (Galaxy 170S, New Brunswick Scientific Co., Inc., Edison, NJ, USA), laser scanning confocal microscopy (SP8, Leica AG, Wetzlar, Germany), transmission electron microscopy (HT7700, Hitachi Co., Ltd., Tokyo, Japan), atomic force microscopy (ARM, Bruker, Ettlingen, Germany), UV spectrophotometer (UV-2700, Shimadzu Corporation, Kyoto, Japan), small animal imager (IVIS^®^Lumina Series III, PerkinElmer, Waltham, MA, USA), flow cytometry (Aria III, BD), and particle size analyzer (Zetasizes Nano ZS, Malvern, UK).

### 2.2. Synthesis of Gold Nanorods

The main factors affecting the synthesis of GNR include: the concentration and type of surfactants, the pH value of the growth solution, and the concentration of the reducing agent. In this chapter, the synthetic method of the GNP mainly refers to the report of Stanislav Emelianov’s team [28]. We thoroughly mixed 0.4 mL of HAuCl_4_(aq) (10 mM), 10 mL of CTAB (aq) (0.1 M), and 22.5 μL of AgNO_3_(aq) (100 mM). Subsequently, we added the 10–30 μL of HCl (1 M) and 525 μL of hydroquinone (1 M) to the growth solution and mixed, and the color of the growth solution changed from orange to clear light yellow. After stirring for 15 min, 10–40 μL of the freshly prepared iced NaBH_4_ (aq) solution was injected into the growth solution by using a micropipette gun. The mixture was stirred at room temperature for 15 s and then placed at room temperature to rest for 16 h. Subsequently, it was centrifuged at 20,000 g for 1 h, the supernatant was absorbed, and the same amount of ddH_2_O was added. These actions repeated twice to stop the growth.

### 2.3. Synthesis of GNR@CTA

The CTA was composed of three independent hairpin probes (AP1, AP2, and AP3), and the 3’ end of each probe is modified with sulfhydryl group. Briefly, 2 μL of AP1 (10 μM), 2 μL of AP2 (10 uM), 2 μL of AP1 (10 μM), and 60 μL of ddH2O were added into 100 μL of EP tube. After annealing at 90 °C for 5 min, the mixture was placed at room temperature for 1 h to complete the self-assembly of CTA. Samples were purified by PAGE and quantified with the Q5000 (Quawell, Fremont, CA, USA). The stabilizer CTAB on the surface of GNR was replaced by CTA modified sulfhydryl groups through ligand exchange. According to the molar ratio of 1:10, fully mixed CTA-SH with GNR. First, the mixture was treated by ultrasound for 5 min, and left to rest for 2 h to fully react. Following the reaction, it was centrifuged at 20,000 g for 1 h, and discarded the supernatant to remove excess CTA. The same amount of ddH_2_O was added, the above-mentioned cleaning process was repeated twice, and the mixture was then re-suspended using ddH_2_O for quantification.

The preparation method, anticancer mechanism, and imaging diagnosis of GNR@CTA are shown in Figure 1. CTA, as a circular ternary aptamer, is realized by precise nucleic acid regulation based on the principle of base complementary pairing. The loading of DOX can be achieved by temperature-regulated CTA denaturation and renaturation. It should be noted that both ends of the CTA have been modified with sulfhydryl groups. Assembly of CTA and GNR by simple mixing and incubation. Due to the strong absorption capacity of GNR in the NIR-II region, GNR can be simultaneously applied in photothermal therapy, Infrared imaging, and PAI imaging. The CTA provides tumor-specific targeting and chemotherapy as targeted warheads and drug depots. This work provides a multifunctional nanoplatform for theragnostic research.

### 2.4. Package of DOX

DOX was mixed with GNR@CTA in a buffer solution containing 0.1 M sodium acetate, 0.05 M NaCl and 5 mM MgCl_2_, and the pH of buffer solution was adjusted to 7.4 before being incubated in a metal bath at a temperature of 37 °C for 1 h with slow shaking. After the incubation, excessive DOX in the supernatant was centrifuged at 3000 g. The conversion of DOX loading capacity is mainly through comparing the change of DOX concentration in buffer solution and supernatant, and detecting the change of fluorescence value at 585 nm under 470 nm excitation by fluorescence spectrophotometer.

### 2.5. In Vitro Cellular Uptake Efficacy of GNR@CTA(DOX)

Cellular uptake can be demonstrated by intracellular DOX fluorescence and FAM-marked CTA. BT474 cells were used as the experimental group, HELF cells were used as the control group. The cells seeded into 24-well plates at a density of 5 × 10^5^ cells per well were incubated with GNR@CTA(DOX) (1 mL, 100 ppm) and DAPI for 0.5 h. After coculturing for 2 h, the 808 nm laser was used to irradiate the cells. After that, the cells were incubated for another 0.5 h to image by confocal laser scanning microscopy (CLSM).

### 2.6. Cell Lines and Culture

Human breast carcinoma cells (BT474 cells) and human embryonic lung fibroblast (HELF cells) were purchased from the cell bank of the Chinese Academy of Sciences. BT474 cells were cultured in RPMI 1640 medium supplemented with 10% FBS and 100 units/mL penicillin/streptomycin and incubated at 37 °C in a humidified incubator containing 5 wt%/vol CO_2_. The HELF cells were under the same culture conditions as the BT474 cells. Specifically, the cell selection was based on the expression of cell surface marker, EpCAM. BT474 cells were used as EpCAM high expression cell group, HELF cells were used as EpCAM low expression cell group.

### 2.7. Cell Viability of GNR@CTA

In order to evaluate the chemo-photothermal synergistic treatment efficiency of the GNR@CTA(DOX) nanocomposites, BT474 cells (10,000 per well) were seeded on 96-well plates and grown in 5% CO_2_ at 37 °C overnight. Then, PBS, GNR@CTA, and GNR@CTA(DOX) were added to the medium. The cells were incubated in 5% CO_2_ at 37 °C for 4 h. The concentrations of DOX were 1, 5, 10, 20, 40, and 80 μg·mL^−1^, respectively. The medium containing samples were washed out and the photothermal treatment was irradiated under 0.8 W·cm^−2^ 1064 nm laser for 5 min. After that, the cells were incubated for 24 h. After incubation, the medium was removed. A volume of 10 µL of MTT solution was added into each well. The cells were cultured for another 4 h. Finally, the supernatant was aspirated. A volume of 150 μL of dimethyl sulfoxide (DMSO) was added to each well. The plate was shaken for 10 min and examined using a microplate reader (Tecan Infinite 1200Pro, Hombrechtikon, Switzerland) at the wavelength of 490 nm.

### 2.8. Animals and In Vivo Study

Equivalent female and male Balb/c nude mice of 6–8 weeks’ old were obtained from Slac Animal Company (Shanghai, China). The mice experiment was approved by the Institutional Animal Care and Use Committee of Anhui Medicine University. We tried our best to minimize animals’ pain and suffering.

The right hind leg of each nude mice was injected with BT474 cells subcutaneously to establish tumors. The tumors were allowed to grow for 7–10 days to reach a size of around 80–100 mm^3^. Then, the BT474 tumor-bearing Balb/c nude mice were randomly divided into 5 groups (n = 3), which were intravenously injected with PBS, GNR@CTA, GNR@CTA (DOX), GNR@CTA+NIR, GNR@CTA (DOX) + NIR, respectively. The injected DOX dose in 100 μL of PBS was 6 mg·kg^−1^ body weight in total. The laser irradiation conditions were 0.8 W·cm^−2^ for 10 min. The body weights and tumor volumes were monitored every other day. The tumor volumes were calculated using the following equation: tumor volume (V) = π × length × width 2/8 × 4/3. Moreover, the tumor tissues (n = 3) in each group were harvested from mice 24 h after the first treatment were also dissected, and fixed in paraformaldehyde used for hematoxylin and eosin (H&E) staining assay. After treatment for 13 days, the mice of each group were randomly chosen and euthanized to retrieve organs (including heart, liver, spleen, lung, and kidney). These excised organs were washed with deionized water and then fixed with 4% formalin solution to stain with H&E.

### 2.9. In Vivo Photothermal Imaging and Photoacoustic Imaging

For in vivo photothermal imaging, suspension of BT474 tumor cells was subcutaneously inoculated into the left armpit of female Balb/c nude mice. When the tumor size reached to around 100–150 mm^3^. The mice were intravenously injected with PBS and GNR@CTA (100 µL, 20 mg·kg^−1^). After that, for 12 h, the tumors were irradiated with 808 nm laser at 0.8 W·cm^−2^ for 5 min, respectively. The photothermal images at different time intervals were recorded by an infrared thermal imaging camera.

For in vivo Photoacoustic imaging, suspension of BT474 tumor cells was subcutaneously inoculated into the left armpit of female Balb/c nude mice. When the tumor size reached to around 100–150 mm^3^. The nude mice were intravenously injected with PBS and GNR@CTA (100 µL, 20 mg·kg^−1^). After that, for 12 h, the nude mice were anesthetized with 2% isofluorane throughout the experiments, and placed in a horizontal position in a holder surrounded by a thin polyethylene membrane to prevent direct contact with water, and allowed acoustic coupling between mouse and transducer array. The photoacoustic imaging images were acquired to cover the subcutaneous tumor regions.

### 2.10. PAGE Assay

DYY-6C electrophoresis analyzer (Liuyi Instrument Company, Shanghai, China) and Bio-rad ChemDoc XRS (Bio-Rad, Hercules, CA, USA) were used to perform 12% Native-PAGE and visualize gel lanes, respectively. Briefly, 15 mL of 12% PAGE composition includes: 6 mL of 30% polyacrylamide, 3 mL of 5× Tris/borate/EDTA (TBE), 5.9 mL of ddH_2_O, 0.1 mL of 10% APS, and 0.01 mL of TEMED. Polyacrylamide gel electrophoresis (PAGE) of the all the experiments required in 0.5× TBE buffer at 80 V for 1.5 h. Preparation of electrophoresis samples: 7 μL of the sample, 2 μL of 6× loading buffer, and 1 μL of SYBR Green I.

## 3. Results and Discussion

### 3.1. Construction and Characterization of GNR@CTA

The preparation method, anticancer mechanism and imaging diagnosis of GNR@CTA are shown in Figure 1. CTA, as a circular ternary aptamer, is realized by precise nucleic acid regulation based on the principle of base complementary pairing. The loading of DOX can be achieved by temperature-regulated CTA denaturation and renaturation. It should be noted that both ends of the CTA have been modified with sulfhydryl groups. Assembly of CTA and GNR by simple mixing and incubation. Due to the strong absorption capacity of GNR in the NIR-II region, GNR can be simultaneously applied in photothermal therapy, Infrared imaging, and PAI imaging. The CTA provides tumor-specific targeting and chemotherapy as targeted warheads and drug depots. This work provides a multifunctional nanoplatform for theragnostic research.

Figure 1a shows the schematic diagram of CTA and the sequence of the required nucleic acid probe. As shown in Figure 1b, CTA consists of three parts: DNA connection chain, drug intercalation arm and aptamer, and we characterized the assembly process of CTA by gel electrophoresis. As show as Figure 1c, AP3 acts as a bridge in the CTA assembly process, AP2 and AP1 are combined with AP3 to form the final assembly. It is worth noting that the three 3’ ends of CTA were modified with sulfhydryl groups. On one hand, it can effectively reduce the degradation of CTA in serum; on the other hand, it can be fixed on the surface of GNR through the Au–S bond. Herein, we mainly characterized the synthesis of GNR in detail. After adjusting the concentration of the surfactant, pH value of the growth solution and the concentration of the reducing agent, the size of GNR synthesized under the optimal concentration condition was uniform. The enlarged TEM image shows that the size of the GNR is about 60–70 nm (Figure 1d). In addition, we calculated the long diameter of the GNR in the TEM image and constructed the long diameter distribution map of the GNR, which is shown in Figure 1e. It can be seen that the average length of GNR is 60 ± 5 nm, the average width is 8 ± 2 nm, and the aspect ratio is 7.5. On this basis, we tested the absorption of GNR by ultraviolet spectrometer. From the data in Figure 1f, the position of the maximum absorption peak of GNR is 1050 nm, which completely meets the requirements of the NIR II window (1000–1350 nm).

### 3.2. Loading and Release of DOX

In this chapter, we use DOX, a classic anti-cancer drug, as an example to analyze the drug loading and release capacity of GNR@CTA. The principle of GNR@ CTA containing DOX is based on the large number of G-C base pairs in CTA. When the G-C base pairs are complementary, DOX can be inserted between the double strands and exist stably, and the fluorescence of DOX will be quenched at the same time. When the laser is irradiated, the GNR photothermal effect promotes the melting of DNA double-stranded and the release of DOX. At the same time, the fluorescence signal of DOX will be recovered again. According to this mechanism, DOX loading and release can be determined by detecting the fluorescence signal intensity in the solution. As shown by the results in Figure 2a, the loading behavior is greatly affected by the GNR@CTA concentration. We determined the optimal mixing ratio of DOX and GNR@CTA by detecting the fluorescence intensity of the remaining DOX in the system. When the concentration of DOX was fixed at 10 μM, we used different concentrations of GNR@CTA to bind DOX in the system. When the concentration of GNR@CTA reached 1.3 μM, the fluorescence intensity of the system reached the minimum value and did not decrease any more, which indicated that the binding of DOX reached the maximum value. Based on the above results, we believe that the optimal mixing concentration ratio of DOX and GNR@CTA is 10:1.3. From the analysis of spectral data, it is difficult for DOX to be 100% binding. Under the optimal ratio, the binding efficiency was about 95%. Moreover, the release behavior is greatly affected by irradiation time. When the 808 laser (600 mW) irradiated to the solution (DOX of 10 μM for 1.3 μM of GNP@CTA), the fluorescence intensity reaches the highest value at 10 min and is equivalent to the fluorescence intensity of the initial DOX solution. This may indicate that the DOX has been completely released at GNP@CTA (1.3 μM) and laser irradiation time (10 min).

In addition, we drew Figure 2b according to the molar ratio. From the figure, it can be concluded that when the molar ratio is 0.13 (that is, the concentration of DOX is 8 times that of GNR@CTA), the load of DOX reaches more than 90% and the saturation does not increase anymore. After determining the appropriate loading concentration ratio, we analyzed the stability of GNR@CTA loaded DOX, and the results are shown in Figure 2c. With DOX as a control, the release rate of DOX in the control group was set to be 100%. When the time was 0 h, DOX was loaded with GNR@CTA, and then placed at room temperature. The fluorescence signal of the incubation solution was detected every 2 h. From the results, the release rate of DOX did not exceed 5% after 10 h of standing, which fully indicated that GNR@CTA(DOX) has superior stability without significant DOX leakage under unstimulated conditions, which fully indicated that GNR@CTA(DOX) has superior stability without significant DOX leakage under unstimulated conditions.

### 3.3. Intracellular Drug Delivery and In Vivo Imaging of GNR@CTA (DOX)

The process of cell-targeted delivery of GNR@CTA(DOX) was analyzed by confocal laser scanning microscopy (CLSM). BT474 cells were used as target cells and HELF as control cells. The GNR@CTA(DOX) nanoparticles can effectively enter the cells through the typical endocytosis process [29]. As presented in Figure 3a, the HELF as the control group had almost no obvious fluorescence, while the surface of the target cell BT474 could be observed with obvious green fluorescence of GNR@CTA (DOX) and weak red fluorescence of DOX. This fully shows that has selective specific targeting, and the weak red fluorescence is mainly due to a little DOX releasing of GNR@CTA (DOX) in acidic conditions of cells. The high selectivity of GNR@CTA (DOX) is based on the fact that CTA contains aptamers that specifically bind to EpCAM. Each CTA structure contains three independent aptamers, which greatly ensures that GNR@CTA (DOX) targeting ability. The sequence of the aptamer was confirmed by previous reports [30].

Furthermore, the carriers promoting the DOX release within the BT474 cells was explored. After DOX or GNR@CTA(DOX) coculturing with BT474 for 2 h, the DOX uptake in the cell reached the maximum at about 90 min and the uptake rate of DOX in simply DOX was about 45%, while GNR@CTA(DOX) was about 81% (Appendix A). This fully indicated that applying GNR@CTA further enhanced the intracellular delivery of chemotherapeutic drugs, thereby promotes synergistic therapy against target cells.

In addition, we also analyzed the retention time of the chemotherapeutic drug DOX after cell uptake, which is closely related to the treatment and drug resistance of target cells. The results are shown in Appendix A. Similarly, we used the fluorescence signal of the DOX stock solution as a control, set the fluorescence intensity to 100%, and set the uptake time and observation time after uptake of the two groups of target cells. In the DOX-only group, after the target cells ingested DOX, the retained DOX in the cells were reduced by 40–50% with about 2 h of culture, and almost all DOX were metabolized after 12 h of culture. In contrast, in the GNR@CTA (DOX) group without NIR irradiation, the retained DOX in the cells were reduced by 10–12% with about 2 h of culture, and reduced by 25–33% after 12 h of culture. This fully indicates that GNR@CTA, as a drug carrier, can significantly improve the enrichment and long-term retention of DOX in target cells. On one hand, it significantly improves the therapeutic effect on target cells, and on the other hand, it is beneficial to reduce drug resistance of cell.

On the basis of the light-triggered temperature rise, we used the infrared thermal imager to record temperature change to study the infrared imaging effect of GNR@CTA. First, we injected GNR@CTA into nude mice through tail vein injection. After 12 h of enrichment, they were irradiated with the 808 laser (600 mW) for different periods of time and then photographed and recorded. As shown in Figure 3b, the temperature of the GNR@CTA group reached above 50 °C after exposing to 808 nm laser for 5 min, whereas the temperature of control group merely raised up to 30 °C under the same condition. Based on the strong near-infrared absorption capacity of GNR@CTA, we tried to verify the potential of GNR@ CTA in vivo photoacoustic imaging (PAI Imaging). The PAI signal of nude mice with breast cancer xenograft tumors was detected after intravenous injection with GNR@CTA. As shown in Figure 3c, after GNR@CTA injection at about 8 h, the tumor site showed a strong PAI signal. At 12 h, the PAI signal of the tumor site reached its maximum value. The above results indicate that GNR@CTA can be used as an enhanced contrast agent.

### 3.4. In Vitro Cytotoxicity of GNR@CTA (DOX)

MTT assay, as a routine cytotoxicity assay, has been used to monitor the tumor-killing effect of various materials. Among them, PBS was used as the control group, GNR@CTA as the material control group, GNR@CTA+NIR as photothermal therapy group, GNR@CTA(DOX) as chemotherapy group and GNR@CTA(DOX) + NIR as photothermal and chemotherapy synergistic therapy group. As Appendix A displayed, consistent with the results described above, GNR@CTA did not show significant cytotoxicity, and the GNR@CTA+NIR group showed a good photothermal treatment effect, and the overall cell death rate exceeded 50% at high concentration. Meanwhile, GNR@CTA(DOX), as a simple drug-loading group, also showed a certain effect on chemotherapy. It is worth noting that the GNR@CTA (DOX) group showed the best inhibition effect than DOX alone and the cell death rate reached more than 90%, which should be attributed to the precise targeted delivery of GNR@CTA.

### 3.5. Application in Tumor Metastasis Model of GNR@CTA (DOX)

In order to further investigate the tumor treatment effect of GNR@CTA (DOX), we constructed the nude mice model of xenograft tumor of the BT474 cell line. After subcutaneously injected BT474 cells into their right dorsal thigh, the tumor-bearing mice were employed to evaluate the tumor suppression efficacy. Forty similar body weight tumor-bearing mice were randomly separated into five groups (n = 8) and then tail vein injected with PBS, GNR@CTA, GNR@CTA(DOX), GNR@CTA + NIR, and GNR@CTA(DOX) + NIR. Twelve hours after injection, the mice in different groups were administrated with or without 808 nm laser irradiation.

In addition, we recorded the weight of nude mice every other day. Figure 4a,b shows the changes of the average tumor volume and average body weight in nude mice over time of treatment. We can see that the average tumor volume of the GNR@CTA(DOX) + NIR group was significantly smaller than other groups and tumor growth was completely inhibited, which fully demonstrates the highly synergistic anti-tumor effect of this system. Moreover, the tumor volume growth rate of the GNR@CTA(DOX) group and GNR@CTA + NIR group decrease in varying degrees compared with PBS, indicating that the simple photothermal treatment and chemotherapy also have certain tumor inhibition effect. Furthermore, the efficacy of GNR@CTA + NIR group was significantly better than chemotherapy alone. The weight of the nude mice did not change during the entire treatment period, the results confirm that the physical signs and status of the nude mice did not change too much during the entire treatment process, eliminating the interference of external influencing factors. After 15 days of observation, all nude mice were sacrificed and the changes of various tumors were recorded by photographing. As shown in Figure 4c, the GNR@CTA(DOX) + NIR group had almost no tumors, and the nude mice were basically cured. In contrast, the PBS group and the GNR@CTA group had the largest tumors, and the tumors began to rot due to oversize tumors, which could endanger the lives of nude mice at any time. However, the tumors in GNR@CTA(DOX) group and GNR@CTA + NIR group grew slowly, there was obviously a limited amount of inhibition. The in vitro images of nude mice tumor shown in Appendix A also prove the above conclusion. H&E Staining results show that in Figure 4d,e, after the photothermal-chemotherapy, the tumor tissue appeared massive cell death, and the originally tight tumor tissue became loose. The results also confirm the effect of synergistic treatment. The pathomorphology analysis results of the main organs are compared between control and GNR@CTA(DOX) + NIR. There was no obvious difference of biochemical and homological parameters in these two groups. The results demonstrate that GNR@CTA(DOX) + NIR has high synergistic anti-tumor effect and biological safety in vivo.

## 4. Discussion and Conclusions

In summary, based on CTA and gold nanorods (GNR), we constructed a near-infrared region II responsive GNR@CTA drug delivery system, which realizes the integration of photothermal, chemotherapy, and imaging of tumor cells. Generally speaking, the optical response to the NIR-II window can be achieved only when the length of the GNR is more than 120 nm. In this chapter, by adjusting the aspect ratio of GNR when the length of GNR does not exceed 80 nm, the span of the optical response from NIR-I to NIR-II is achieved. The smaller size of nano-particle gives GNR better photo-thermal stability, deeper tissue penetration and higher temporal-spatial resolution. Based on GNR, we successfully prepared a novel GNR@CTA nanocarrier with high anticancer drug delivery capability, superior NIR-II response photothermal conversion capability and good biocompatibility by self-assembly with CTA. In addition, we realized that the combined treatment of photothermo-chemotherapy has a significant synergistic effect on inhibiting tumor growth through GNR@CTA loading DOX. At the same time, the results of infrared/photoacoustic imaging also indicate that GNR@CTA in the treatment of imaging monitoring also shows its superiority. In this study, the design idea of GNR and the construction of drug-loading platform further expand the application of gold nanoparticles in the NIR-II window, and also provide a new strategy for the design of DNA-based nano-drug delivery systems.

## Data Availability

Not applicable.

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
