# Peer review of "High Performance Gold Nanorods@DNA Self-Assembled Drug-Loading System for Cancer Thermo-Chemotherapy in the Second Near-Infrared Optical Window"

_pharmaceutics, 2022, doi:10.3390/pharmaceutics14051110_

Round 1

Reviewer 1 Report

This manuscript aimed to develop and characterize the complexes of gold nanorods with DNA-doxorubicin conjugates as novel drug delivery systems in oncology. Although much effort has been put in performing these experiments and some significant results were obtained, there are several MAJOR concerns in the manuscript that need to be considered and clarified.

Major concerns include:

  1. ‘Methods’ section.

One critical aspect of publishing research is describing the methods used in enough detail that the experiments can be reproduced by others. However, this is not the case here. Generally, the text is sloppy and important information is missing. In the section 2.5. ‘NIR-responsive drug release in vivo and in vitro’, the authors are not writing only about the drug release, but also about determination of tumor-suppressing activity, histological studies etc. The authors only mention studies in BT474 cells and later in the text we see the results also in control HELF cells. It should be noted that the authors did not even mention that what kind of cells they used, that BT474 are breast cancer cells and HELF cells are human embryonic lung fibroblasts. Why did you choose it as a control and is it comparable with breast cancer cells? Some other methodological concerns will be mentioned later. I would add here that there is no information on Ethical approval to perform this study in this section.

  1. Lack of Discussion.

There is no discussion of obtained results at all, not even one reference. However, there are studies with similar approach (of course not completely the same) than can be compared with the results of this study.

  1. Lack of Supplementary material.

Although authors write about some results that are presented elsewhere, I didn’t find any supplementary material in the journal electronic submission system.

  1. Language.

As the major concern I would also emphasize the quality of English language in the manuscript. It requires detailed language editing since it contains too many grammatical and spelling errors. Some examples are:

Line 28: ‘This work enables to obtaining a stimuliresponsive…’

Line 39: ‘Therefore, more effective cancer treatment exists great significance.’

Line 65: ‘…because for in vivo applications.’

Line 70: ‘The new gold nanorods has…’

I have several other specific comments/suggestions:

Line 22 – the term ‘encapsulated’ is not completely appropriate

Line 57 – please be more consistent if you use the term ‘gold’ or ‘Au’

Lines 71-72: ‘The ultraviolet absorption spectrum shows that the absorption of this GNR is 1050 nm.’ – UV spectrum is approx. in the range 10-400 nm wavelengths, definitely not above 1000 nm

Materials and Methods: subscripts and superscripts are not applied in the text

Section 2.2. – please add reference

Section 2.3. – The term ‘self-assembly’ was stated in the title. Please explain it and add a reference for the preparation method.

Section 2.5. – generally it should be divided in in vitro and in vivo part and explained in much more details.

Line 136: Why 5 min when it is stated in the Results that the fluorescence intensity reached the highest value at 10 min?

Lines 152-1661: this is not appropriate for the ‘Result’ section – it should be either in Methods or at the end of Discussion as summarizing the effects.

Lines 169-170: gel electrophoresis is not mentioned in Methods.

Line 177: similar for TEM.

Figure 1e: are these results obtained by zeta sizer? if so, please add it in the text.

Figure 1f: UV spectrum and high wavelengths

Line 201: what does ‘concentration time’ mean?

Lines 203-204: not completely clear what you wanted to say, please rephrase it (1.3 uM in comparison to 1.7 and 2 uM)

Line 218: the sentence is not finished!

Line 227: control cells first time mentioned here

Line 232: How do you explain that high selectivity? Are there similar results for GNP or some their complexes in the literature?

Lines 237, 242: you refer to supplementary figures, however they are not uploaded in the system.

Lines 290-292: ‘the tumor-bearing mice were employed to evaluate the tumor suppression efficacy when the tumor reached more than 5*5*5’ and in the Methods it was stated that ‘when the tumor size reaches 3*3*3 or more, GNR@CTA (DOX) is injected intratumorally’

Reviewer 2 Report

A really interesting and thorough study, nice clear diagrams explaining the complex concept. Worthy of publication. One clarification I wanted to make - and perhaps could be explained better in the text - in Figure 2a is the different spectra the amount of drug mixed with the rods, or the definite amount loaded? Is there an assumption that 100% loading occurs? I wonder would this be better displayed in concentration instead of showing the fluorescence (though I understand the point the image makes after irradiation).

Round 2

Reviewer 1 Report

Manuscript is improved after revision.